

# Waveguidability of idealized midlatitude jets and the limitations of ray tracing theory

Volkmar Wirth[1]

[1]Institute for Atmospheric Physics, Johannes Gutenberg University Mainz, Becherweg 21, 55128 Mainz, Germany.

**Correspondence:** Volkmar Wirth (vwirth@uni-mainz.de)

**Abstract.** Ray paths of stationary Rossby waves emanating from a local mid-latitude source are usually refracted equatorward. However, this general tendency for equatorward propagation is mitigated by the presence of a midlatitude jet which acts as a zonal waveguide. This opens the possibility for circum-global teleconnections and quasi-resonance, which suggests that the ability to guide a wave in the zonal direction is an important jet property. This paper investigates waveguidability of idealized midlatitude jets in a barotropic model on the sphere. A forced-dissipative model configuration with a local source for Rossby waves is used in order to quantify waveguidability by diagnosing the latitudinal distribution of waviness in a longitudinal sector far downstream of the forcing. Systematic sensitivity experiments show that waveguidability increases smoothly with increasing jet amplitude and with decreasing jet width. This result is contrasted with the predictions from two idealized theoretical concepts based on (1) ray tracing as derived from Wenztel-Kramers-Brillouin (short: WKB) theory and (2) a sharp jet with a zonally oriented front of potential vorticity. The existence of two so-called turning latitudes, which is the key diagnostic for a zonal waveguide according to ray tracing theory, turns out to be a poor predictor for the dependence of waveguidability on jet amplitude and jet width obtained in the numerical simulations. By contrast, the meridional gradient of potential vorticity correlates fairly well with the diagnosed waveguidability. The poor predictions from ray tracing are not surprising, because the underlying WKB assumptions are not satisfied in the current context. The failure of WKB is traced back to the properties of the underlying equations, and a heuristic argument is presented to elucidate the potential of the PV gradient to act as a proxy for waveguidability.

## 1 Introduction

Rossby waves are an ubiquitous feature of the atmospheric flow in the upper troposphere (Rossby, 1940; Rhines, 2002). They can transfer energy and momentum across large distances and sometimes give rise to teleconnections (Wallace and Gutzler, 1981; Hoskins and Karoly, 1981; Branstator, 2002). The present paper focuses on Rossby waves in midlatitudes, where they may occur in the form of Rossby wave packets (Wirth et al., 2018).





An important aspect in connection with midlatitude Rossby waves is the extent to which they are ducted in the zonal direction. As is well known, there is a general tendency for Rossby waves to be refracted equatorward owing to the sphericity of
the Earth (Hoskins and Karoly, 1981; Hoskins and Ambrizzi, 1993); in practice, this holds true to the extent that the background flow varies smoothly. However, the existence of a zonal jet may change the situation and lead to preferential propagation in the zonal direction. The latter situation is often refered to as a zonal waveguide (Branstator, 2002; Schwierz et al., 2004; Martius et al., 2010; Branstator and Teng, 2017). An interesting question is what aspect of the background atmosphere constitutes a strong zonal waveguide. Earlier work of Manola et al. (2013) suggests that strong and narrow jets are associated with high
waveguidabilty, although their work does not suggest a mechanistic explanation.

   To the extent that the midlatitude background flow represents an efficient waveguide, this may lead to circumglobal Rossby waves, which in turn can result in circumglobal teleconnections (Branstator, 2002; Ding and Wang, 2005; Feldstein and Dayan, 2008; O'Reilly et al., 2018). If, in addition, such circumglobal Rossby waves turn quasi-stationary, this has two implications: first, weather is synchronized across long distances (Kornhuber et al., 2019), and, second, there is the possibility of con-
structive interference of the Rossby wave with itself, which may lead to quasi-resonance in the case of stationary forcing (Petoukhov et al., 2013; Kornhuber et al., 2017). In fact, a number of previous authors argued that an increased tendency for quasi-resonance may be responsible for the increased occurrence of recent Northern Hemisphere weather extremes (Petoukhov et al., 2013; Coumou et al., 2014; Petoukhov et al., 2016; Stadtherr et al., 2016; Kornhuber et al., 2016; Mann et al., 2017). The issue is interesting because quasi-resonance would imply that small changes in the properties of the jet and its associ-
ated waveguidability can entail large changes in Rossby wave amplitude. This scenario provides additional motivation for a comprehesive understanding of waveguidability.

   A key argument in the work of previous authors to support the theory of quasi-resonance draws on the ideas of ray tracing, which in turn are based on the Wenztel-Kramers-Brillouin (short: WKB) theory (Lighthill, 1967). Ray tracing allows one to diagnose the refraction of ray path along which the waves propagate. In this theory, a zonal jet turns into an efficient waveguide
as soon as a region of wavelike propagation in the neighborhood of the jet is straddled by two so-called turning latitudes on either side of the jet, where a suitably defined "refractive index" turns zero. Not surprisingly, the existence of two turning latitudes plays a crucial role in the analysis of the above quoted papers in their attempt to connect the observations with the theory of quasi-resonance (e.g., Fig. 4 in Petoukhov et al. 2013, Fig. 2 in Petoukhov et al. 2016, Fig. 1 in Kornhuber et al. 2016). Note that the predictions of ray tracing are based on properties of the background flow alone (like the refractive index),
i.e., they do not imply any explicit reference to the Rossby waves.

   The basic tenet of WKB theory is that the scale $\Delta_w$ of the wave must be much smaller than the scale $\Delta_{bg}$ on which the background flow varies, i.e.,

$$\frac{\Delta_w}{\Delta_{bg}} \ll 1 \,. \tag{1}$$

Unfortunately, this assumption is often violated in connection with Rossby waves. For instance, in the analysis of Kornhuber
et al. (2016), the two turning latitudes are separated by about $10° \approx 1000$ km (see their Fig. 1), which corresponds to $\Delta_{bg} \approx$ 330 km; but at the same time the wavelength of the waves in question (zonal wavenumber 7 at midlatitudes) is on the order of





4000 km, which means that $\Delta_w \approx 650$ km. In this situation, the relation (1) is badly violated and one would not expect WKB theory to be applicable at all.

This state of affairs motivates the current work, in which we consider the waveguidability of midlatitudes jets in an idealized
modeling framework and investigate the validity of ray tracing arguments. Our approach is partly based on the work of Manola et al. (2013), but with modifications and extensions in a number of ways. Key to our analysis is a forced-dissipative model configuration, which allows us to explictly simulate the propagation of stationary Rossby waves emanating from a local source. To the extent that the path of propagation is confined in the zonal direction, the scenario will be associated with a high degree of waveguidability. These results will be compared with the predictions from ray tracing theory. We will see that the existence
of two turning latitudes is a poor predictor for waveguidability. Instead, the strength of the meridional gradient of potential vorticity turns out to be a better proxy for waveguidability in our framework. In addition, we will corroborate our results by analysing the underlying equations, which is facilitated by the idealized nature of our model.

The paper is organized as follows. First in section 2 we present our model, its configuration, and our method of numerical solution. Section 3 reviews the relevant theoretical concepts, before our results are presented and compared with the theoretical
concepts in section 4. Section 5 seeks a deeper understanding of the numerical results through reference to the linearized equations, and finally section 6 gives a short summary and our conclusions.

## 2 Model configuration and numerical solution

We consider non-divergent barotropic flow on a sphere in a forced-dissipative configuration. In our analysis we focus on the stationary part of the solution, which is obtained through temporal averaging. Key model variable is absolute vorticity $q$, which
plays the role of a potential vorticity (short: PV) in the barotropic model. It is given by

$$q(\lambda, \phi, t) = 2\Omega \sin\phi + \zeta \,, \tag{2}$$

where

$$\zeta(\lambda, \phi, t) = \frac{1}{a\cos\phi}\frac{\partial v}{\partial\lambda} - \frac{1}{a\cos\phi}\frac{\partial}{\partial\phi}(u\cos\phi) \,, \tag{3}$$

is relative vorticity, $\mathbf{v} = (u, v)$ is the horizontal wind, $\lambda$ is longitude, $\phi$ is latitude, $t$ is time, $a$ is the Earth's radius, and $\Omega$ is the
angular velocity of the Earth's rotation. The dynamics are determined by

$$\frac{Dq}{Dt} = -\lambda_r(q - q_0) + F \,, \tag{4}$$

where $D/Dt = \partial/\partial t + \mathbf{v}\cdot\nabla$ denotes the material derivative, $\lambda_r$ is a damping rate,

$$q_0(\phi) = 2\Omega\sin\phi - \frac{1}{a\cos\phi}\frac{d}{d\phi}(u_0\cos\phi) \tag{5}$$

is a zonally symmetric background PV (with $u_0(\phi)$ denoting the corresponding background zonal wind), and $F$ represents the
forcing. For later reference we note that

$$\frac{1}{a}\frac{dq_0}{d\phi} = \beta(\phi) - \frac{d}{a\,d\phi}\left[\frac{1}{\cos\phi}\frac{d}{a\,d\phi}(u_0\cos\phi)\right] \tag{6}$$





with

$$\beta(\phi) = \frac{2\Omega}{a} \cos\phi .$$ (7)

The forcing is implemented as pseudo-orographic forcing with a rather local orography, i.e.,

$F = div\left(f_0 \mathbf{v}_f h\right)$ (8)

with $f_0 = 10^{-4} \mathrm{s}^{-1}$, $\mathbf{v}_f = (u_f, 0)$,

$$u_f(\phi) = U_f \cos\phi$$ (9)

(using $U_f = 15 \ \mathrm{m\,s^{-1}}$), and with a Gaussian-shaped orography

$$h(\lambda, \phi) = h_0 \, \exp\left(-\frac{(\lambda - \lambda_0)^2}{2\sigma_\lambda^2} - \frac{(\phi - \phi_0)^2}{2\sigma_\phi^2}\right) ,$$ (10)

where $h_0 = 0.3$, $\sigma_\lambda = 10°$, $\sigma_\phi = 10°$, $\lambda_0 = 30°$E, and $\phi_0 = 45°$N. Our formulation (8) for the forcing has the advantage that it integrates to zero upon global integration, which means that it creates equal amounts of positive and negative local PV anomalies (in contrast with the forcing used by Manola et al. 2013). Note also that our forcing $F$ contains a fixed and specified flow field $\mathbf{v}_f$ and is, therefore, independent of the actual flow $\mathbf{v}$. The forcing is localized in the sense that the spatial scale of the pseudo-orography $h$ is much smaller than the planetary scale.

The dissipative term in (4) is designed such that in the unforced case ($F = 0$) the background state $q_0(\phi)$ is a stationary solution of the equation. Waves, i.e., deviations from zonal symmetry, arise to the extent that $F$ is nonzero. But even for non-zero forcing the solution is gradually relaxed back towards the zonally symmetric background state. As relaxation parameter we use $\lambda_r = (7 \, \mathrm{days})^{-1}$, which was chosen such that Rossby wave packets emanating from the pseudo-orography are just about able to propagate once around the Earth before they get completely dissipated. This facilitates the interpretation of our 105   solutions because it deliberately avoids constructive interference of a Rossby wave train with itself and, hence, the phenomenon of quasi-resonance. It contrasts our approach from that of Manola et al. (2013), who used a slightly smaller value for $\lambda_r$ and obtained weak resonance phenomena.

Any deviation from purely zonal flow will be considered to be an eddy or, more specifically, a wave. Correspondingly, we will use maps of the meridional wind $v$ in order to visualize the waves.

The flow is initialized with the zonally symmetric background state. The latter is defined by specifying a zonally symmetric zonal flow field $u_0(\phi)$ and computing the corresponding $q_0(\phi)$ via (2). Our main strategy includes a set of sensitivity experiments in which we impose different background flows $u_0(\phi)$ and investigate their impact on the stationary part of the solution. More specifically, the background flow is defined as a superposition of a solid-body rotation $u_{SB}(\phi)$, a mid-latitude jet $U_J(\phi)$, and a small linear component $L(\phi)$, i.e.,

$u_0(\phi) = u_{\mathrm{SB}}(\phi) + u_J(\phi) + L(\phi) .$ (11)



The solid body rotation part is given by

$$u_{\text{SB}}(\phi) = U_{\text{SB}} \cos\phi \tag{12}$$

with $U_{\text{SB}} = 15$ m s$^{-1}$, and the jet part is given by a Gaussian profile

$$u_J(\phi) = U_J \, e^{-\frac{(\phi-\phi_0)^2}{2\sigma_J^2}} \tag{13}$$

with $\phi_0 = 45\,°$N . The latter choice is motivated by the work of Manola et al. (2013), who showed that a Gaussian profile fits the shape of observed jets fairly well. The two parameters $U_J$ and $\sigma_J$ characterize the amplitude and width of the jet, respectively. The value of $U_J$ will be varied between 0 and 40 m s$^{-1}$. For the width parameter we choose $\sigma = 5°$ unless stated otherwise, and we will refer to this choice as a "narrow" jet. Generally, the sum of $u_{SB}(\phi)$ and $u_J(\phi)$ would be non-zero at the poles, which is undesirable. We, therefore, included the linear function $L(\phi) = a + b\phi$ on the right hand side of (11) with

the parameters $a$ and $b$ chosen such that the resulting total profile $u_0(\phi)$ turns zero at both poles. Quite deliberately we choose our background wind to be positive for all latitudes in order to allow stationary wave propagation. As explained below, this enables us to diagnose the "waveguidability" of the jet in a meaningful manner. For later reference we note that

$$\left.\frac{d^2 u_J}{d\phi^2}\right|_{\phi=\phi_J} = -\frac{U_J}{\sigma_J^2} \; . \tag{14}$$

    Numerical solutions are obtained using a standard pseudo-spectral scheme. To ensure numerical stability, we add a hyper-

diffusion term

$$T_h = D \, \nabla^4 (q - q_0) \tag{15}$$

to the right hand side of (4) with $D = 10^{15} \text{m}^4 \text{s}^{-2}$. The term $T_h$ is generally small in comparison with the other terms in (4) and will be neglected in our theoretical analysis. The code uses triangular truncation at wavenumber $N = 72$. Time-stepping is done using the leapfrog scheme with a time step of $\Delta t = 20$ minutes, combined with a Robert-Asselin filter in order to damp

the computational mode.

    For some choices of the backgroud wind our numerical solutions turn out to be highly transient. This transience is due to barotropic instability in the case of a strong narrow jet. However, in all our experiments these unstable modes are characterized by a large eastward phase velocity. This is illustrated in Fig. 1 showing a Hovmoeller diagram of the transient part of the meridional wind, $v' = v - \overline{v}$, where the overbar denotes the time average. The Hovmoeller diagram indicates how barotropic

instability is triggered downstream of the orography (which is located at $\lambda_0 = 30°$). During the first 10 days the instability materializes in the form of an eastward propagating Rossby wave packet with non-zero (eastward) phase speed. Later this wave packet disperses in the zonal direction, and after about day 30 it has turned into a almost pure sine wave with eastward phase velocity. This behavior allows us to eliminate the transient part of the solution by simple time averaging. More specifically, for each model run we performed a 100-day long model integration and extracted the stationary part of the flow by averaging over

the last 90 days (the initial 10 days were discarded in order to eliminate initial transients).

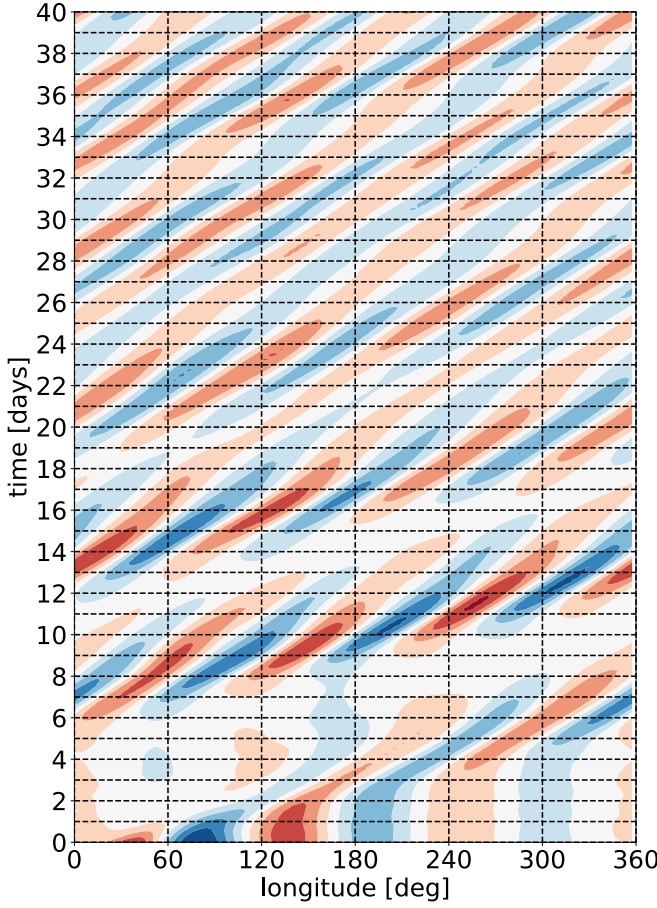

**Figure 1.** Development of a barotropically unstable initial state for a model set-up with a strong narrow jet at $45°$N. The plot shows a Hovmoeller diagram of the meridional wind component of the transient part of the flow, $v' = v - \overline{v}$, averaged over $30 – 60°$N.

For illustration we consider two very different background flows, one with pure solid body rotation (corresponding to $U_J = 0$ m s$^{-1}$) and one with a strong narrow jet superimposed (corresponding to $U_J = 40$ m s$^{-1}$). The two latitudinal profiles $U_J(\phi)$ are depicted in Fig. 2. The corresponding numerical solutions for $\overline{v}$ are presented in Fig. 3. In the case of the solid-body rotation, the transients have died out after about 10 days such that the solution for $v$ at later times (not shown) is practically

indistinguishable from the time average $\overline{v}$ displayed in Fig. 3a. By contrast, in the strong jet case there is strong transience throughout the integration, but these transients are effectively eliminated by the time averaging leaving us with the forced stationary part $\overline{v}$ displayed in Fig. 3b.

Apparently, for pure solid body rotation (Fig. 3a) there is a wavetrain emanating from the pseudo-orography; downstream of the source the individual troughs and ridges develop a strong NE-SW tilt and the wavetrain crosses the equator and disperses

into the Southern Hemisphere (Hoskins et al., 1977; Hoskins and Karoly, 1981). At the same time, the wave signal is damped

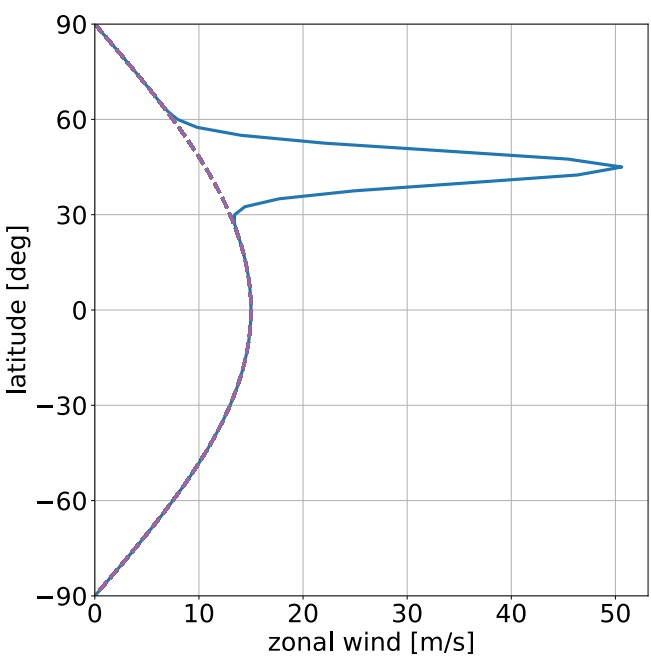

**Figure 2.** Latitudinal profiles of the background zonal wind $u_0(\phi)$ for the strong narrow jet case (solid line) and for the pure solid body rotation case (dashed line).

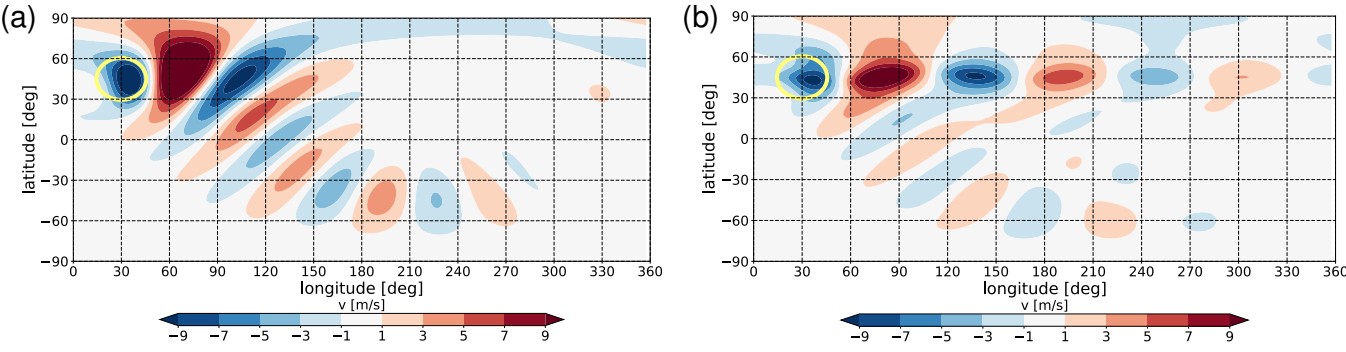

**Figure 3.** Illustration of the numerical solution for an initial state with (a) pure solid body rotation, and (b) a strong narrow jet sumperimposed on solid body rotation. Displayed in both cases is the time averaged meridional wind $\overline{v}$. The oval-shaped yellow contour is the contour of the orography $h(\lambda, \phi)$ at $0.3\,h_0$.





with increasing distance from the wave source, which is due to our relaxation term in (4). The cross-equatorial propagation implies that some 180° downstream of the Rossby wave source the wavetrain is found in the Southern Hemisphere and the Northern Hemisphere is practically void of any wave signal at these longitudes.

By contrast, the solution for the strong jet case (Fig. 3b) indicates that the majority of the wave signal remains in the Northern Hemisphere midlatitudes, and only a rather small fraction of the wave signal follows a great circle into the Southern Hemisphere. This behavior is consistent with the notion that a strong narrow jet acts as a waveguide (Manola et al., 2013). As a consequence, in this case the majority of the wave signal 180° downstream of the Rossby wave source has remained in the Northern Hemisphere midlatitudes. Later in section 4.1 we will use the striking difference in behavior between these two contrasting scenarios to define a quantiative measure for "waveguidability".

## 3   Theoretical concepts

Before we do so, however, we review in this section some well-known concepts for the analysis of stationary Rossby waves and their propagation in a spherical domain. This will help us to interpret our results in the later parts of the paper.

First, we linearize the equation of motion about the background state $q_0(\phi)$ and obtain the following equation for the (small) perturbation $q' = q - q_0$,

$$\left(\frac{\partial}{\partial t} + u_0 \frac{\partial}{a\cos\phi\,\partial\lambda}\right) q' + v' \frac{dq_0}{a\,d\phi} = S' \, , \tag{16}$$

where

$$S' = -\lambda_r q' + F \, . \tag{17}$$

Note that the forcing $F$ is now assumed to be small and, hence, a perturbation term. Expressing, as usually (see, e.g., Vallis 2006), the perturbation variables $q'$ and $v'$ in terms of a perturbation streamfunction $\psi'$, this becomes

$$\left(\frac{\partial}{\partial t} + u_0 \frac{\partial}{a\cos\phi\,\partial\lambda}\right) \nabla^2 \psi' + \frac{\partial\psi'}{a\cos\phi\,\partial\lambda} \frac{dq_0}{a\,d\phi} = S' \, , \tag{18}$$

where $\nabla^2$ is the two-dimensional Laplace operator in spherical coordinates.

For further progress is turns out convenient to perform a coordinate transformation corresponding to a Mercator projection of the sphere onto a plane (see Hoskins and Karoly 1981; Hoskins and Ambrizzi 1993). This is achieved by defining

$$x \;=\; a\lambda \, , \tag{19}$$
$$y \;=\; a\ln\left(\frac{1+\sin\phi}{\cos\phi}\right) \, . \tag{20}$$

This transformation is valid everywhere except at the poles; it transforms (18) into

$$\left(\frac{\partial}{\partial t} + u_M \frac{\partial}{\partial x}\right) \left(\frac{\partial^2\psi'}{\partial x^2} + \frac{\partial^2\psi'}{\partial y^2}\right) + \beta_M \frac{\partial\psi'}{\partial x} = S' \, , \tag{21}$$





where

$$u_M = \frac{u_0}{\cos \phi} \tag{22}$$

and

$$\beta_M = \cos \phi \, \frac{dq_0}{a \, d\phi} \equiv \frac{dq_0}{dy} \; . \tag{23}$$

Assuming $u_M$ and $\beta_M$ to be constants, one can look for plane wave solutions of the form

$$\psi'(x, y, t) = \hat{\psi} e^{i(kx + ly - \omega t)} \; , \tag{24}$$

where $k$ and $l$ are the wavenumbers in the zonal and meridional direction on the Mercator projection, respectively, and $\omega$ is the

angular velocity. Note that $k$ relates to the dimensionless zonal wavenumber $s$ through

$$s = k \, a \; . \tag{25}$$

Perturbations of the form (24) are solutions of (21) if the following dispersion relation is satisfied

$$\omega = u_M k - \frac{k \beta_M}{k^2 + l^2} \; . \tag{26}$$

Obviously, the advantage of using the new coordinates $(x, y)$ is that both the equation (21) and the associated dispersion relation

(26) are formally identical to their Cartesian $\beta$-plane version, if the zonal basic flow $u_0$ is replaced by $u_M$ and the planetary vorticity gradient $\beta$ is replaced by $\beta_M$.

For later reference we note that for a background flow with pure solid body rotation as in (12) one obtains $u_M = U_{\mathrm{SB}} = const$ and

$$\beta_M = 2 \left( \Omega + \frac{U_{\mathrm{SB}}}{a} \right) \frac{\cos^2 \phi}{a} \; . \tag{27}$$

The latter is obviously not a constant, which, hence, violates the assumption made above when deriving the dispersion relation.

### 3.1 WKB approximation and ray tracing

If the factors $u_M(\phi)$ and $\beta_M(\phi)$ in (21) depend on latitude instead of being constants, further analytical progress is possible by assuming that the spatial variation of $u_M$ and $\beta_M$ is weak in the sense that their derivatives can be neglected in comparison with the rate of change resulting from the wave's phase. This method is often referred to as WKB approximation (see, e.g.,

Lighthill 1967). In this framework, a plane wave of the form (24) can still be considered to be an appoximate solution of (21). For the special case of stationary waves, this immediately leads to

$$k^2 + l^2 = \frac{\beta_M}{u_M} \tag{28}$$

or, using (25),

$$a^2 l^2 = \hat{K}_s^2 - s^2 \; , \tag{29}$$

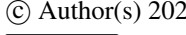



where the square of the dimensionless stationary wavenumber is defined as

$$\hat{K}_s^2 := a^2 \frac{\beta_M}{u_M} \ . \tag{30}$$

Note that $\hat{K}_s^2$ inherits the property of "weak variation" from $u_M$ and $\beta_M$. Apparently, for a given zonal wavenumber $s$ the solution (24) is wavelike in those regions where the meridional wavenumber $l^2 > 0$. In these regions the Rossby wave packets are free to propagate. For illustration, Fig. 4 shows profiles of $u_0$, $\beta_M$, and $\hat{K}_s = (\hat{K}_s^2)^{1/2}$ for pure solid body rotation (top

row), for the strong narrow jet (bottom row), and for a weaker jet (middle row) representing a scenario intermediate between the former two. The profiles of $\hat{K}_s$ in Fig. 4 indicate that for solid body rotation wave propagation in the WKB sense is possible over most of the sphere except in the regions close to the poles. On the other hand, for background flows with a jet, both flanks of the jet are characterized by a strip of relatively low (Fig. 4f) or even imaginary (Fig. 4i) values of $\hat{K}_s$; depending on the zonal wavenumber, this potentially prohibits wave propagation in these strips.

Ray tracing arguments consider the propagation of Rossby waves in those regions which allow wave propagation (e.g., Hoskins and Karoly 1981; Hoskins and Ambrizzi 1993). It can be shown that the vector of the group velocity is refracted towards larger values of $\hat{K}_s^2$ as one proceeds in the direction of the propagation; in addition, latitudes where $l = 0$ are associated with a purely zonal group velocity, which implies that ray paths return back to latitudes where the wave packet came from. The latter is only valid as long as the background state does not vanish at those latitudes, which is satisfied for all background states

used in our study. Latitudes where $l(\phi)$ turns zero are aptly called "turning latitudes".

Turning latitudes can easily be diagnosed from profiles of $\hat{K}_s$ as those latitudes where the function $\hat{K}_s(\phi)$ intersects the straight line $s = const$. Let us, for illustration, consider a wave with $s = 4$. Figure 4c then indicates that for solid body rotation there are two turning, one at 60°N and one at 60°S . This is consistent with our numerical solution from Fig. 3a which shows wavelike behaviour throughout most of the domain. By contrast, both cases with a jet superimposed (Figs. 4f and 4i) indicate

that waves emanating from a source at 45°N encounter two turning latitudes, one at 50°N and another one at 35°N . We note in passing that the strong jet case displayed in the bottom row of Fig. 4 features regions with a negative meridional gradient of absolute vorticity and, hence, of $\beta_M$ (Fig. 4h). This opens the possibility for barotropic instability (Charney and Stern, 1962), which we do observe in our numerical simulations.

The jet scenario is particularly interesting in the present context. Whenever there are two turning latitudes ($l = 0$) separated

by a region of wavelike propagation ($l > 0$), ray tracing arguments predict that Rossby waves packets oscillate between the two turning latitudes as they propagate eastward. This means that they are effectively trapped between the two turning latitudes and, thus, ducted in the zonal direction. In other words, ray tracing theory suggests that the existence of two turning latitudes is tantamount to a perfect zonal waveguide. In fact, this seems to be broadly consistent with our numerical solution shown in Fig. 3b, where the majority of the wave signal is ducted in the zonal direction: the corresponding function $\hat{K}_s(\phi)$ shows the

existence of two turning latitudes for any wavenumber $1 \le s \le 7$, stradeling a wavelike region in the core of the jet.

Let us return to the case of a background flow with pure solid body rotation according to (12), yielding

$$\hat{K}_s^2(\phi) = 2 \left( \frac{\Omega a}{U_{\mathrm{SB}}} + 1 \right) \cos^2 \phi \tag{31}$$



**Figure 4.** Diagnosing the zonally symmetric background state for three different cases: solid body rotation (top row), a weak narrow jet with $U_J = 10$ m s$^{-1}$ (middle row), and a strong narrow jet with $U_J = 40$ m s$^{-1}$ (bottom row). In each case the left column shows the meridional wind profile $u_0(\phi)$, the middle column shows $\beta_M(\phi)$, and the right column shows the dimensionless stationary wavenumber $\hat{K}_s(\phi)$, where the negative values (shading) represent minus the imaginary part of $\hat{K}_s$.





(cf. Fig. 4c). For this case, ray tracing theory predicts that ray paths are generally refracted equatorward, because the function $\hat{K}_s^2(\phi)$ maximizes right at the equator. In fact, Hoskins and Karoly (1981) showed explicitly that ray paths on a solid body

rotation background flow are identical to great circles. In addition, $\hat{K}_s^2$ goes to zero at both pole such that at some point the expression on the right hand side of (29) turns negative leading to a turning latitude close to the pole. The crucial question in our context is whether or not there is a second (more equatorward) turning latitude which prevents the general equatorward refraction and forces the ray path back into the zonal direction. The latter scenario would be equivalent to a perfect zonal waveguide. It is this prediction from ray tracing theory that led previous authors to systematically search for the occurrence

of two turning latitudes stradeling a region of wave propagation. Interestingly, this prediction is not fully supported by our numerical simulations, as we will show in section 4.

### 3.2 PV front

Another school of thought associates a midlatitude zonal waveguide with the existence of a sharp meridional gradient of PV (Schwierz et al., 2004; Martius et al., 2010). A highly idealized model that represents this idea would be a zonally oriented front

of PV in the basic state separating two regions with a completely homogeneous PV distribution (Platzmann, 1949). Note that the idealization made in this model is opposite to that in the WKB approximation: a PV front is equivalent to a discontinuous jump in the background PV, while the WKB approximation assumes the background state to vary very gently.

 The background flow associated with a PV discontinuity is a westerly jet with a cusp-like peak at the latitude of the discontinuity. Solutions of the linearized equations can be found which are wavelike in the zonal direction and evanescent in the

meridional direction like

$$\psi' = \hat{\psi} e^{-sgn(y-y_0)k(y-y_0)} e^{i(kx-\omega t)} , \tag{32}$$

where we assumed Cartesian geometry for simplicity, where $sgn(....)$ denotes the sign-function, and where the other symbols have their usual meaning. The resulting dispersion relation is similar (albeit not identical) to the dispersion relation of Rossby waves on a $\beta$ plane with constant background flow (see, e.g., Schwierz et al. 2004); in particular, both kinds of waves

have westward phase propagation with respect to the basic flow. More importantly, however, these interfacial waves can only propagate in the zonal direction and their amplitude is expontially damped away from the PV front. This is consistent with the concept of the key importance of PV gradients for the existence of Rossby waves. It transpires that in this model a zonal waveguide is present to the extent that the PV distribution has a zonally oriented sharp PV gradient separating two regions with near-homogeneous PV on both sides of the sharp gradient. As we will see, our numerical solutions turn out to be broadly

consistent with this idea.





## 4 Investigating waveguidability

Based on the theoretical concepts sketched in the previous section, we now define "waveguidability" in the framework of our numerical model, explore it systematically for various background flows, and compare the results with predictions from ray tracing theory.

### 4.1 Definition of waveguidability

As we argued before, background solid body rotation can be considered as a reference scenario in which the waves propagate along great circles from one hemisphere to the other. As a consequence, wave activity emanating from a local Northern-Hemispheric Rossby wave source can be expected to be located almost entirely in the Southern Hemisphere some $180°$ downstream (in longitude) of the source region. We interpret this scenario as the lack of waveguidability, and our diagnostic (to be

defined shortly) should reflect this by producing a very small value.

These considerations motivatated us to use the following method to quantify "waveguidability". Introducing wave enstrophy of the stationary part of the solution as

$$\mathcal{E}(\lambda,\phi) = \frac{1}{2}(\overline{q}^*)^2 \tag{33}$$

where $q^* = q - [q]$ is the deviation of absolute vorticity from the zonal mean and the overline denotes the time average, we

define a probability density function as

$$P(\phi) = \frac{\cos\phi}{N} \int_{\lambda_1}^{\lambda_2} \mathcal{E}(\lambda,\phi)\,d\lambda\,, \tag{34}$$

where a suitable downstream sector is defined through $\lambda_1 \leq \lambda \leq \lambda_2$ with $\lambda_1 = 180°$ and $\lambda_2 = 270°$ and where $N$ represents a normalization factor to guarantee that $\int_{-\pi/2}^{\pi/2} P(\phi)\,d\phi = 1$. Note that our definition of $P(\phi)$ includes a factor $\cos\phi$ in order to account for the variation of surface area with latitude. In other words, the function $P(\phi)$ quantifies the likelihood to encounter

wave enstrophy $\mathcal{E}$ as a function of latitude in the downstream sector $[\lambda_1, \lambda_2]$. Fig. 5 shows this probability density function for our two background profiles from Fig. 2. Apparently, for solid body rotation most of the wave enstrophy in the downstream sector is found in the Southern midlatitdes, consistent with Fig. 3a. On the other hand, for the strong narrow jet, a large fraction of the wave enstrophy remains in the Northern midlatitudes, which is consistent with Fig. 3b.

The probability density function $P(\phi)$ is now used to define waveguidability $\mathcal{W}$ as the probability to encounter downstream

wave enstrophy in the northern midlatitudes at $\phi_1 \leq \phi \leq \phi_2$, i.e.,

$$\mathcal{W} = \int_{\phi_1}^{\phi_2} P(\phi')\,d\phi'\,, \tag{35}$$

where we set $\phi_1 = 30\,°\mathrm{N}$ and $\phi_2 = 60\,°\mathrm{N}$. By design, $\mathcal{W}$ is a number between 0 and 1, and it will be expressed in % in the following. For the case of solid body rotation (Fig. 5a), we obtain $\mathcal{W} = 7.5\%$, indicating that this background flow has





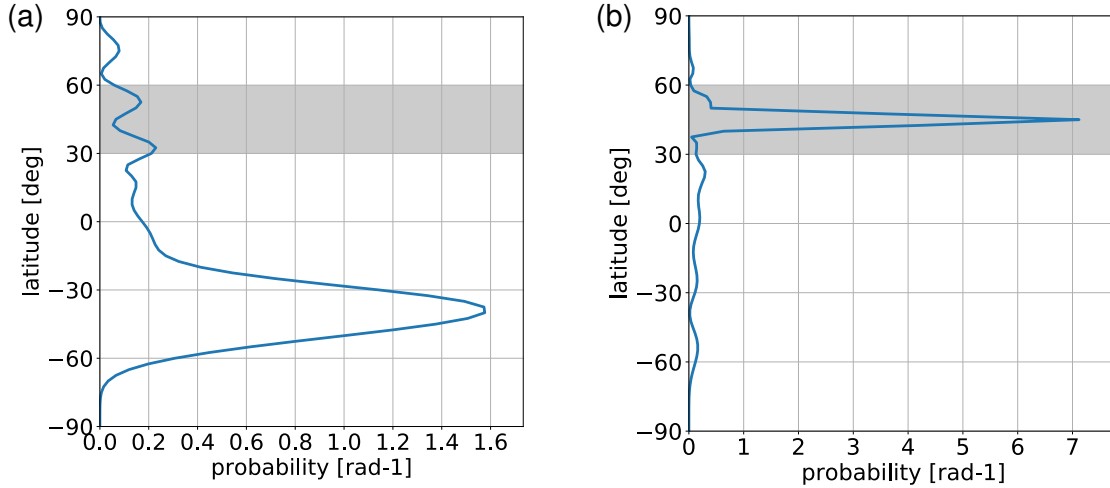

**Figure 5.** Probability density function $P(\phi)$ as defined in (34) for two different background flows: (a) solid body rotation, and (b) a strong narrow jet with $U_J = 40$ m s$^{-1}$. The grey shading indicates the Northern midlatitudes, where the forcing is located. Broadly speaking, the function $P(\phi)$ quantifies where the "wavines" can be found far downstream of the Rossby wave source.

a very low capacity to duct waviness in the zonal direction. On the other hand, for the strong narrow jet (Fig. 5b) we obtain

$\mathcal{W} = 75.4\%$; although this value is still less than 100%, it is an order of magnitude larger than for pure solid body rotation.

Our method to quantify waveguidability is based on, but also extends the ideas from Manola et al. (2013). A key difference is that we diagnose the meridional distribution of waviness only in a downstream longitudinal sector rather than circumglobally, which allows us to associate a percentage (i.e., a number between 0 and 100) to the waveguidability of any jet. Nevertheless, our definition of waveguidability is neither universal nor exact. It is not universal because it only makes sense in our partic-

ular model configuration where we consider the stationary response from a local Rossby wave source in a forced-dissipative framework; it is not exact because it slightly depends on model details like, e.g., the exact value of the damping parameter $\lambda_r$, the choice of the boundaries for the downstream region $[\lambda_1, \lambda_2]$, and the definition of "Northern midlatitudes" $[\phi_1, \phi_2]$. Nevertheless, this diagnostic is appropriate for our purpose, because we will only study relative changes of $\mathcal{W}$ as we change the background flow.

**4.2 Waveguidability as function of jet amplitude and jet width**

We now systematically vary the amplitude $U_J$ and the width $\sigma_J$ of our background jet (13) and use our numerical solutions to compute the value of $\mathcal{W}$ as explained above. The result is shown in Fig. 6 (red squares and thin red line). Apparently, there is a very smooth and gradual variation of $\mathcal{W}$ with both $U_J$ and $\sigma_J$ extending from rather low values around 10% to rather high values close to 80% (within the ranges considered for $U_J$ and $\sigma_J$). Qualitatively the behavior is entirely consistent with

the results of Manola et al. (2013), who suggested increasing waveguidability for increasing jet amplitude and decreasing jet width.





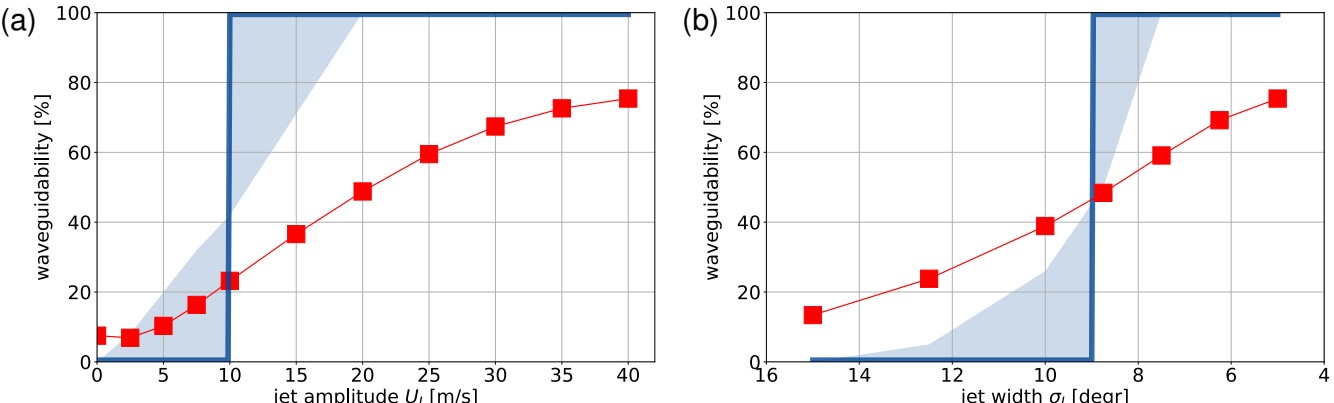

**Figure 6.** Dependence of waveguidability $\mathcal{W}$ on (a) jet strength $U_J$ (with $\sigma_J = 5\,^\circ$ kept fixed), and (b) jet width $\sigma_J$ (with $U_J = 40\ \mathrm{m\,s^{-1}}$ kept fixed). The red squares (connected by a thin red line) represent the values of $\mathcal{W}$ diagnosed from the numerical solutions. The solid blue line represents the prediction from ray tracing theory using zonal wavenumber $s = 4$. The blue shaded area indicates the uncertainty of the ray tracing prediction associated with the fact that the numerical solution does not have a fixed single zonal wavenumber (see text for details).

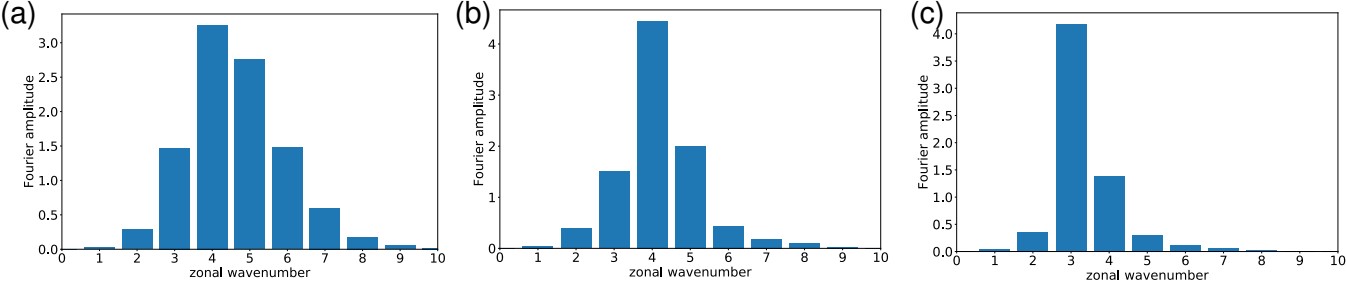

**Figure 7.** Zonal Fourierspectrum of $\overline{v}$ averaged over $30 - 60^\circ$N , for (a) solid body rotation, (b) a weak narrow jet with $U_J = 10\ \mathrm{m\,s^{-1}}$ , and (c) a strong narrow jet with $U_J = 40\ \mathrm{m\,s^{-1}}$ .

### 4.3 Comparision with theoretical expectations

The numerical results from Fig. 6 are now contrasted with the predictions from theoretical concepts that we sketched in the previous section. We start with ray tracing theory, in which the profile of $\hat{K}_s^2$ is the key diagnostic and the existence of two

turning latitudes indicates a perfect waveguide. The difficulty in applying this theory is that it is not clear, a priori, which zonal wavenumber to consider as the relevant one. Our numerical solution indicates generally an entire range of wavenumbers (Fig. 7). For the background profiles with solid body rotation and for weak jets, the Fourier amplitudes maximise at $s = 4$ (Fig. 7a and b), while for the stronger and narrower jets the Fourier amplitudes maximise ad $s = 3$ (Fig. 7c). Taking $s = 4$ as the relevant zonal wavenumber, we checked whether there are two turning latitudes bounding a wavelike region within the jet.

Whenever this turned out to be the case, we associated this jet with the value $\mathcal{W} = 100\%$, otherwise we set $\mathcal{W} = 0\%$. This



literal interpretation of ray tracing theory is depicted as the solid blue line in Fig. 6. It is clear that the resulting discontinuous dependence of $\mathcal{W}$ on $U_J$ and $\sigma_J$ (thick blue line) is in stark contrast with the very gradual behavior obtained in the numerical solutions (red squares and thin red line). In addition, waveguidability diagnosed from the numerical solutions is much less than 100% in a large part of the range of parameters where ray tracing theory predicts a perfect waveguide. For instance, a narrow jet with $U_J \approx 11$ m s$^{-1}$ yields $\mathcal{W} \approx 26\%$ in our simulations — rather than 100% as predicted by ray tracing theory.


One could argue that the above interpretation of ray tracing theory is too brute-force because it does not account for the existence of an entire spectrum of zonal wavenumbers (see Fig. 7). We, therefore, considered another extreme scenario by assuming that all zonal wavenumbers have equal amplitude. The value of $\mathcal{W}$ for this scenario was obtained as the fraction of the number of wavenumbers with two turning latitudes divided by the number of all wavenumbers allowing wavelike behavior within the jet region. The resulting values are represented as the other edge of the blue-shaded area in Fig. 6. Apparently, there is now a more gradual transition from $\mathcal{W} = 0\%$ to $\mathcal{W} = 100\%$ as jet amplitude and jet width are varied, but the transition is still significantly steeper than for the values obtained from the simulations. In addition, the scenario with equal amplitude for all relevant zonal wavenumber is not realistic, either, because even for very weak jets our numerical solutions indicate a maximum spectral amplitude as some intermediate wavenumber. In the end, the best estimate for a "fair" prediction from ray tracing theory is a line which is located somewhere in the middle of the blue shaded area in Fig. 6. This yields a fairly steep (albeit not discontinuous) transition in both panels of the figure, which — again — is in stark contrast with the very gradual curves obtained from the simulations. But even for jets where the ray tracing interpretation is unambiguous, its prediction is still far off the result of our simulations: for the narrow jet with $U_J = 20$ m s$^{-1}$, ray tracing theory unambiguously predicts $\mathcal{W} = 100\%$, while our analysis suggests a much smaller value of $\mathcal{W} \approx 50\%$.



It is enlightening to focus on the bahavior in Fig. 6a for $20\,\text{m s}^{-1} \leq U_J \leq 40$ m s$^{-1}$. Within this range, the numerical solutions indicate an increase of $\mathcal{W}$ from 49% to 75%. On the other hand, the ray tracing prediction is 100% for all those jets, and this prediction is unambiguous because there are two turning latitudes for any zonal wavenumber in question. This behavior is due to the fact that within this range of $U_J$-values the profiles of $\hat{K}_s(\phi)$ are practically independent of $U_J$. This, in turn, is related to the fact that for narrow strong jets the meridional gradient of absolute vorticity is dominated by the meridional curvature of the background wind field. In this case the meridional gradient of background PV (6) can be approximated as



$$\frac{dq_0}{a\,d\phi} \approx -\frac{1}{a^2}\frac{d^2 u_0}{d\phi^2} \tag{36}$$

and, using (22), (23), and (30), one obtains

$$\hat{K}_s^2 \approx -\frac{\cos^2 \phi}{u_0}\frac{d^2 u_0}{d\phi^2}\;. \tag{37}$$

For our Gaussian jet with a fixed width $\sigma_J$, this together with (14) yields

$$\left.\hat{K}_s^2\right|_{\phi_J} \approx +\frac{\cos^2 \phi_J}{\sigma_J^2} \tag{38}$$


at the jet latitude $\phi_J$, which is apparently independent of jet amplitude $U_J$. This argument explains why ray tracing theory is unable to predict the substantial increase in $\mathcal{W}$ when increasing the jet amplitude from 20 to 40 m s$^{-1}$. We conclude that





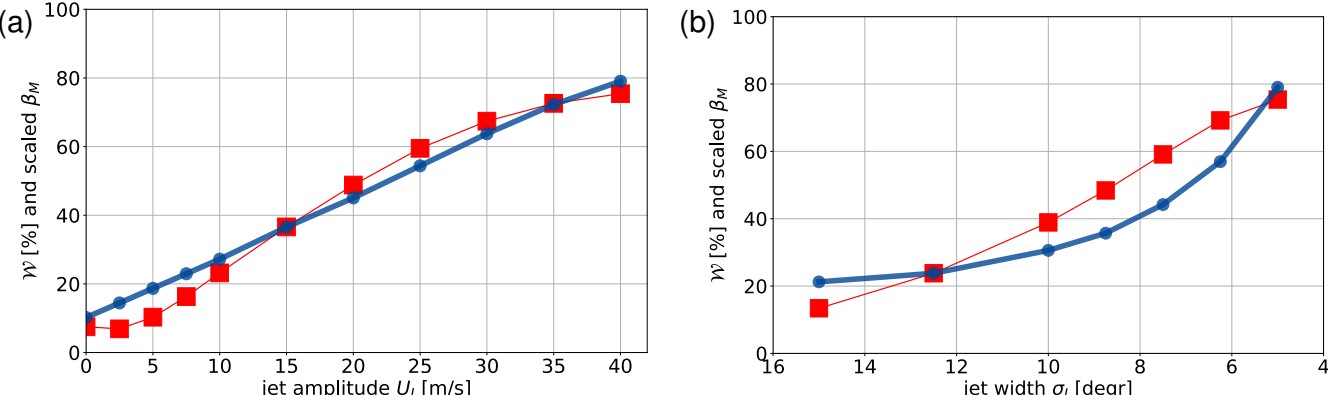

**Figure 8.** Same as Fig. 6, except that here the solid blue line represents the non-dimensionalized $\beta_M$ (see text for details).

ray tracing theory does a poor job in predicting the variability of waveguidability as we vary either the jet amplitude or the jet width.

How does the other theoretical concept fare, in which the meridional gradient of PV plays the key role for waveguidability? To the extent that the PV gradient is a relevant proxy for waveguidability, we expect a close correlation between $\mathcal{W}$ and $\beta_M$ at jet latitude. For this reason, we plot in Fig. 8 the values of $\mathcal{W}$ together with the values of $\beta_M$ at $\phi_J$ after we nondimensionalized $\beta_M$ by multiplying with $f_s = 8.5 \times 10^{11}\,\text{ms}$ (with the same $f_s$ in both panels). Regarding the dependence of $\mathcal{W}$ on jet amplitude $U_J$, the PV gradient (viz. $\beta_M$) turns out to be a very good predictor, showing a smooth increase from small to large values.

In particular, the PV gradient increases for values of $U_J$ in the range $20\,\text{m s}^{-1} \leq U_J \leq 40\,\text{m s}^{-1}$ (unlike the ray tracing prediction), which is also visible in panels e and h of Fig. 4. Regarding the dependence of $\mathcal{W}$ on jet width $\sigma_J$, the PV gradient still predicts a smooth increase from small to large values, although the blue line is less linear than the red line. Note that, at jet latitude $\phi_J$, the dependence of $\beta_M$ on $U_J$ and $\sigma_J$ can be obtained by combining (23) with (6) and using (14). This yields

$$
\begin{aligned}
\beta_M(U_J, \sigma_J) &\approx \frac{2\Omega}{a}\cos^2\phi_J + \frac{\cos\phi_J}{a^2}\frac{U_J}{\sigma_J^2} \\
&= A + B\,\frac{U_J}{\sigma_J^2}\,.
\end{aligned}
\tag{39}
$$


with positive constants $A$ and $B$, which explains the linear increase of $\beta_M$ with increasing $U_J$ in Fig. 8a and the smooth but nonlinear decrease of $\beta_M$ with increasing $\sigma_J$ in Fig. 8b.

  Overall, a comparison between Figs. 6 and 8 shows that, in our model experiments, waveguidability correlates much better with the meridional PV gradient than with the predictions from ray tracing theory. This result is synthesized in Fig. 9, which

shows the results of a systematic parameter sweep varying $\sigma_J$ between 5 and 15° in steps of 2.5° and varying $U_J$ between 0 and $40\,\text{m s}^{-1}$ in steps of $5\,\text{m s}^{-1}$. Apparently, the variation of $\mathcal{W}$ and $\beta_M$ in this parameter space is qualitatively similar.





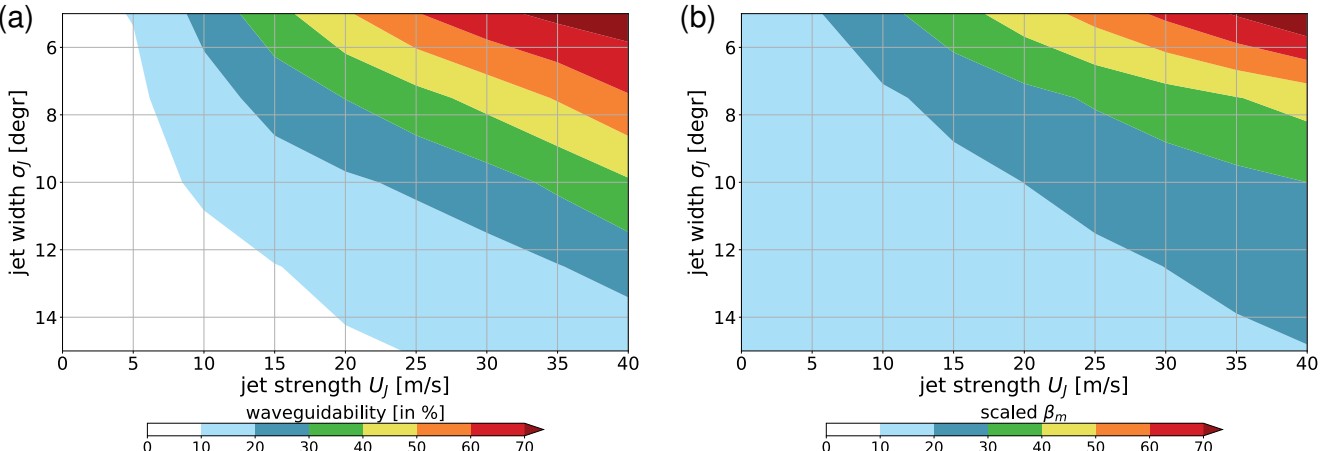

**Figure 9.** Results from the parameter sweep involving $5 \times 9$ model simulations with 5 different values for $\sigma_J$ and 9 different values for $U_J$: (a) waveguidability $\mathcal{W}$, (b) scaled $\beta_M$ (see text for details).

## 5 Analysis of the linearized equations

We now aim to obtain a deeper understanding for why WKB-based ray tracing theory provides partly misleading results regarding waveguidability. For this purpose we linearize the model equation (16) about the background state and restrict

attention to stationary flow, yielding

$$\frac{u_0}{a\cos\phi}\frac{\partial q'}{\partial\lambda} + \frac{v'}{a}\frac{dq_0}{d\phi} = -\lambda_r q' + F \,. \tag{40}$$

Note that this equation does not imply any weak-variation assumptions, which means that the coefficients $u_0$ and $q_0$ may be arbitrary functions of latitude. This means that (40) can be applied to a broader class of problems than covered by WKB theory.

We, first, investigate whether or not our numerical solutions are close to linear. For this purpose we repeated the simulations

with the forcing amplitude $h_0$ reduced by a factor $10^4$ and the resulting perturbation fields multiplied by a factor $10^4$. This so-called pseudo-linear solution is shown in Fig. 10b and compared with the full nonlinear solution shown in Fig. 10a. As it turns out, both solutions are very similar, especially regarding their global appearance. There are only small local differences in the Northern midlatitudes, where the pseudo-linear solution is somewhat more peaked, suggesting that the effect of the nonlinear terms in the equation is similar to weak diffusion. Yet, the overall character of the solution is well captured by linear dynamics.

We now write the perturbation PV in terms of the perturbation streamfunction,

$$q' = \nabla^2 \psi' \,. \tag{41}$$

The zonal periodicity of our domain allows a Fourier series expansion of both $F$ and $\psi'$ like

$$F = \sum_{s=0}^{\infty} \hat{F}(s,\phi)\, e^{is\lambda} \,, \quad \psi' = \sum_{s=0}^{\infty} \hat{\psi}(s,\phi)\, e^{is\lambda} \,. \tag{42}$$



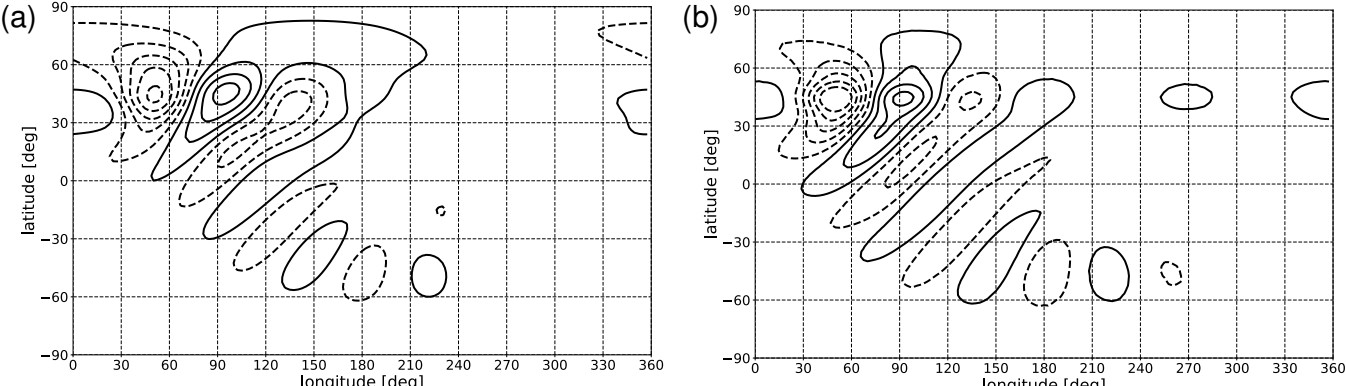

**Figure 10.** Numerical solution of the stationary part $\overline{\psi}^*$ of the perturbation streamfunction for the weak narrow jet with $u_J = 10$ m s$^{-1}$: (a) full nonlinear solution, and (b) pseudo-linear solution. The contours are chosen to be symmetric about zero and the contour interval is $0.4 \times 10^7$ m$^2$s$^{-1}$ (positive contours solid, negative contours dashed).

This yields the following equation for the (possibly complex) Fourier coefficients

$$\frac{(1 - i\tilde{\epsilon})}{\cos\phi} \frac{\partial}{\partial\phi} \left( \cos\phi \frac{\partial\hat{\psi}}{\partial\phi} \right) + \left[ \tilde{K}_s^2 - \frac{(1 - i\tilde{\epsilon})\,s^2}{\cos^2\phi} \right] \hat{\psi} = -i \frac{a^3 \cos\phi}{s u_0} \hat{F} \,, \tag{43}$$

with the two dimensionless parameters

$$\tilde{\epsilon} = a\cos\phi \frac{\lambda_r}{s\,u_0} \tag{44}$$

and

$$\tilde{K}_s^2 = a^2 \frac{\frac{1}{a}\frac{dq_0}{d\phi}}{u_0} \,. \tag{45}$$

Note that $\tilde{K}_s^2$ and $\hat{K}_s^2$ are related through $\tilde{K}_s^2 \cos^2\phi = \hat{K}_s^2$. Equation (43) is an inhomogeneous Helmholtz-like equation, and the appropriate boundary condition is $\hat{\psi} = 0$ at both poles. The term in angle brackets, called $T(\phi)$, may be either positive or negative. For positive $T$, the character of the (homogeneous part of the) solution $\hat{\psi}(\phi)$ is oscillatory, while for negative $T$ the character of the solution $\hat{\psi}(\phi)$ is exponentially evanescent. Our profiles of $\hat{K}_s^2(\phi)$ shown in Fig. 4 indicate that the solution has an oscillatory character for solid body rotation or for a weak jet, as long as the zonal wavenumber $s$ is small enough. However, in the case of a strong jet there are small regions with evanescent behavior on both sides of the jet where $\hat{K}^2$ is negative. In either case, the solution $\hat{\psi}$ at a given latitude does not only depend on the local coefficients $\tilde{\epsilon}$ and $\hat{K}_s^2$ at that latitude, but it may "feel" the values of these coefficients at other latitudes. In particular, the solution can "tunnel through" regions where $T < 0$, if that region is small enough and if there is another oscillatory region beyond. This is in stark contrast with ray tracing theory, which is purely local in the sense that the direction of the rath path and the amplitude of the solution are given by the local properties of the basic state. The ray path solution only needs two turning latitudes in order to be completely confined





in the meridional direction, resulting in a perfect zonal waveguide. By contrast, the mathematical character of (43) indicates that there may be situations in which tray tracing gives misleading results. In particular, the possibility of "tunneling" through finite regions of exponential behavior may explain why our numerical solutions show a weak wave signal in the Southern Hemisphere even for the strong narrow jet (Fig. 3b)

The other question we want to discuss here is why the PV gradient within the jet is possibly a more appropriate proxy for its waveguidability. Let us, for the moment, consider the unforced, undamped version of (43), which reads

$$\frac{1}{\cos\phi}\frac{\partial}{\partial\phi}\left(\cos\phi\frac{\partial\hat{\psi}}{\partial\phi}\right) + \left(\tilde{K}_s^2 - \frac{s^2}{\cos^2\phi}\right)\hat{\psi} = 0\,. \tag{46}$$

For weak damping, solutions of the full problem (43) should *locally* be close to solutions of (46). Interestingly, the entire information about the background atmosphere in the latter equation is contained in the dimensionless parameter $\tilde{K}_s^2 = \hat{K}_s^2/\cos^2\phi$.

It implies that the solution of this equation can only depend on the combination of background fields contained in $\hat{K}_s^2$. This is in conflict with our numerical results from section 4.2, which show a better correlation of waveguidability with $\beta_M$ than with $\hat{K}_s$. The only possible conclusion is that our results about waveguidability and their dependence on the background flow are essentially determined by the forced-dissipative nature of our model configuration.

The remaining question in this line of argument is why the *global* character of the solution should depend on dissipation in

an essential manner. At this point we suggest a heuristic argument according to which the global character of the solution is a result of a competition between the zonal speed of Rossby wave propagation (in the group velocity sense) and the dissipation. While the rate of dissipation is constant and fixed (being equal to $\lambda_r$), the speed of zonal propagation may depend, inter alia, on the meridional gradient of PV. The latter idea would, indeed, be consistent with the concept of downstream development in the framework of PV thinking (see Fig. 8 in Wirth et al. 2018 for details): a given PV anomaly at the leading edge of an Rossby

wave packet induces a new PV anomaly on the downstream side, and the speed of downstream propagation should be related to the strength of the meridional PV gradient. In competition against the omni-present dissipation, those parts of the Rossby wave field that have a larger propagation speed can travel longer distances before they are dissipated. In a sense, regions with large PV gradients are like "express lanes" for Rossby wave packets. On the other hand, those parts of the wave that deviate from the "express lane" soon find themselves in regions with a much smaller meridional PV gradient, implying much slower

zonal propagation. Since these wave packets are subject to the same damping rate, they can travel only much smaller distances before they are dissipated. It follows that $180°$ downstream of our local forcing only those wave packets that used the "express lane" have survived significant damping, and this might explain why waveguidability as defined in section 4.1 is large for jets with large $\beta_M$.

## 6 Summary, discussion, and conclusions

In this paper we investigated the waveguidability of idealized midlatitude jets using a barotropic model for flow on a sphere. Our focus was on stationary waves in a forced-dissipative model configuration with a local source for Rossby waves. This set-up allowed us to quantify waveguidability (in percent) by diagnosing the latitudinal distribution of the explicitly simulated





wave enstrophy in a longitudinal sector far downstream of the forcing. We carried out a systematic sensitivity study by varying the amplitude and the width of the jet.

Our numerical solutions indicate that waveguidability increases smoothly with increasing jet amplitude and with decreasing jet width. This result is in contrast to the prediction from WKB-based ray tracing theory, where the waveguide is either perfect or non-existent, depending on the existence or non-existence of two turning latitudes (one on either side of the jet). Another weakness of ray tracing theory is that the profile of the all-important stationary wavenumber saturates at some point for fairly narrow and strong jets; this theory would, therefore, not predict any further increase of waveguidability as the jet's amplitude is

further increased, which is in conflict with the simulated behavior. To be sure, the overall poor predictive power of ray tracing theory in the current application is not surprising because the underlying WKB assumptions are not satisfied. We conclude that a literal application of ray tracing theory can be misleading; in particular, the existence of two turning latitudes is by no means a binary event at which the waveguidability suddenly changes in a fundamental way.

Some In addition, we found that waveguidability in our simulations correlates much better with the meridional gradient of back-

groud PV at jet latitude than with the predictions from ray tracing theory: both the meridional PV gradient and waveguidability vary smoothly as a function of jet amplitude and jet width. This result is consistent with the idealized concept of Rossby waves on a zonally oriented PV front suggesting that sharp meridional gradients of PV are conducive to zonal Rossby wave propagation. The good correlation with the background PV gradient was argued to be due to the forced-dissipative character of our model set-up. A heuristic argument was presented based on the idea that the reach of a Rossby wave packet is determined by a

balance between the dissipation rate and the speed of propagation of the Rossby wave packet; to the extent that the zonal speed of propagation is proportional to the meridional PV gradient, waveguidability in the zonal direction should, indeed, be related to the PV gradient.

Analysis of the linearized model equation (but without the WKB approximation) indicates that the amplitude of the forced stationary solution is governed by an inhomogeneous one-dimensional Helmholtz equation. This implies that the solution

at a given grid point depends on the spatial distribution of the coefficients in a non-local fashion. Furthermore, a wave-like solution can tunnel through small regions of evanescent behavior, which means that the existence of two turning latitudes is not necessarily equivalent to a perfect waveguide. By contrast, the ray path solution only depends on the local values of the coefficients which prevents the possibility of tunneling. This argument helps to understand why the literal interpretation of ray tracing theory can be misleading and why the existence of two turning latitudes is often associated with values of

waveguidability well below 100%.

Overall, our analysis is not very affirmative regarding the utility of ray tracing for diagnosing waveguidability of Rossby waves in midlatitude jets. This seems to be in conflict with results from earlier publications such as Hoskins and Ambrizzi (1993), who extended the WKB concept to longitudinally varying background flows and showed that the spatial distribution of the stationary wavenumber has a certain amount of predictive power. Later, it was even argued that this kind of analysis

may be useful in explaining the occurrence of persistent regimes and climate extremes (Hoskins and Woollings, 2015). The point is that these applications mostly rely on a broad qualitative interpretation of ray tracing. In fact, a comparison of the global distribution of the fields of $\hat{K}_s$ and $\beta_M$ in Hoskins and Ambrizzi (1993, their Fig. 3) indicates that both fields suggest

broadly the same behavior regarding the horizontal propagation of Rossby waves on the sphere. In other words, as long as the interpretation does not focus on the details of the diagnosed fields, it may well be qualitatively consistent with the observed

behavior. What we have shown here is that the literal application of ray tracing and, in particular, the search for two turning latitudes as indication for a perfect waveguide must be considered with care.

Despite these caveats, an analysis that focuses on the existence of two turning latitudes in the background flow may still have some merit. As suggested by our Fig. 6, such a binary definition of events distinguishes background states with smaller values of $\mathcal{W}$ from background states with larger values of $\mathcal{W}$. In a statistical analysis which — from a large sample — distinguishes

episodes with and without two turning latitudes, the sample with two turning latitudes will, on average, have larger waveguidability than the complementary sample without two turning latitudes. To the extent that the sample with two turning latitudes is associated with specific properties which the complementary sample is lacking, these properties are likely to have a systematic association with waveguidability.

On the more constructive side, our work suggests that the meridional PV gradient might be a suitable proxy for waveguid-

ability. Obviously, this hypothesis is currently lacking any underlying closed theory. In addition, it should be tested in more realistic modeling frameworks, because our barotropic model entirely lacks the partitioning of PV into a contribution due to vorticity and a contribution due to static stability. By contrast, Ertel PV in a primitive equation model would be able to capture the sharp gradients of PV on isentropes at the boundary between the upper troposphere and the lower stratosphere (Martius et al., 2010). It is left for future work to explore suitable diagnostics for waveguidability in such more realistic frameworks.

*Code availability.* The computer code is available from the author upon request

*Author contributions.* The author did all the work related to this paper by himself.

*Competing interests.* The author declares that he has no conflict of interest.

*Acknowledgements.* Part of the research leading to these results has been done within the subproject "Upscale impact of diabatic processes from convective to near-hemispheric scale" of the Transregional Collaborative Research Center SFB/TRR 165 "Waves to Weather"
(www.wavestoweather.de) funded by the German Research Foundation (DFG).





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
