# Peer review of "Waveguidability of idealized midlatitude jets and the limitations of ray tracing theory"

_Weather and Climate Dynamics, 2020_

## Referee Comment (RC1) · Olivia Romppainen-Martius (Referee) · 5 Feb 2020

This paper presents a systematic analysis of the sensitivity of Rossby wave propagation to the strength and width of the extratropical jet stream. A simple but convincing set of idealized simulations are run to explore the role of turning latitudes and PV gradients for wave propagability. It is a joy to read this well organized and clear paper. Two theoretical concepts are introduced and compared against the model results. The limitations of the model simulations are discussed. I recommend acceptance after minor revisions.

Minor points: L2 which → that L28 suggest to replace what aspect with which properties L35 This is an aspect of the QR theory that is also unclear to me. In my understand-

ing an interference of the wave with itself is not needed for QR. Rather the interference with the forced signal is crucial. However, if the wave is circumpolar the interference should be constructive. See also: https://link.springer.com/article/10.1007/s40641-019-00150-x L50 This statement is unclear for me. L54ff This is slightly confusing since one scale is in the meridional direction and the other scale in the longitudinal direction. L91 why is vf set to zero? L90ff A schematic depiction of the forcing could help the reader. L137 delete however L158 void → devoid L168 could you add the equation of motion for the convenience of the reader? L203 weak → small L217 ON the other hand → in contrast L220 which → that L227 turning latitudes L292 On the other hand → in contrast L310 How independent are the jet width and the jet strength? Can these two parameters be varied completely independently? L345 behavior L405 This can also interpreted from a PV anomaly point of view. The far field effect of PV anomalies would allow waves to cross areas of weaker gradients and "reinitate" a wave on a downstream waveguide. L424 maybe add the equation for the group velocity.

---

## Referee Comment (RC2) · Tim Woollings (Referee) · 17 Feb 2020

This is a very nice paper which provides useful analysis and insight into the concept of waveguiding. I suggest acceptance subject to minor revisions.

One general query I had is whether any comment can be made on how much the results might apply to non-stationary situations. The analysis focuses on stationary waves but it seems the mechanisms should be similar? The O'Reilly et al case is an interesting one in this regard, with an initial transient development with non-zero phase speed acting to set up the 'circumglobal' stationary pattern.

I wonder whether the discussion on ray tracing should be broadened a little. Of course there are many issues with a literal interpretation of this, as the author shows very con-

vincingly, but the results here are quite specific to a zonal waveguiding situation. More broadly, ray tracing theory has some predictive power to suggest wave paths resulting from vorticity sources, for example, which just a quantification of the PV gradient doesn't. (Though this application also has limitations, and should in general not be relied upon in isolation.) Essentially, the theory also suggests a refraction towards higher values of K_s^2, not just the simplistic turning latitude approach taken here.

My only real concern with the paper (and a minor one) is the use of just one measure of waves, named epsilon here (eq 33). In some applications, the approach is to define a conservative wave activity, by dividing the enstrophy by the mean PV gradient. This is probably not important, but I just wonder if this would alter some of the interpretation of K_s - ie could K_s have a better relationship to the waveguidability W if a different measure of wave activity was used?

Minor queries

- line 35: Does quasi-resonance concern the resonance of the wave with itself, as suggested here, or the resonance of forced and free waves?

- line 50: not sure this is right. In ray tracing you have to specify the zonal wavenumber, which is a property of the wave.

- line 55: clarify the length scales in this paragraph

- I didn't find fig 1 very convincing. It seems there is a lot of transient activity which has to be averaged out, though I suppose this is not unlike reality. Are the magnitudes in fig 1 weaker than those in fig 3b?

- lines 246-251: These sentences seem a bit out of place in a paragraph on solid body rotation.

- The focus is exclusively on zonal PV gradients - can the concept be extended to relate to meridional gradients and wave propagation.

- l267-9: I'm not sure this means much - surely this is the only thing that can happen in that model?

- l306: I would have thought it depends strongly, not slightly, on the choice of downstream longitudes.

- I found the argument around line 420 a bit hard to follow. Presumably a key step is that the linearisation can be excluded as a reason, given fig 10?

- the discussion at the end of section 5 is interesting. So would the author expect $K_s$ to be more useful in the limit of no damping? Are the experiments sensitive to the damping?

typos - l45: straddled - search for this - l59: midlatitude - l74: The key ? - l245: poles - l266: exponentially - l292: midlatitudes

---

## Author Comment (AC1) · 4 Mar 2020

I am grateful to both reviewers for carefully reading the manuscript and making suggestions for improvements. In this document I explain my response to all individual comments as well as the changes I made to the manusscript. The reviewers' comments are repeated in black font, followed by my replies and the explanations how I revised the manuscript in blue font. In addition I provide all changes to the manuscript in the form of output from the latexdiff command.

**Reviewer 1 (Olivia Martius)**

This paper presents a systematic analysis of the sensitivity of Rossby wave propagation to the strength and width of the extratropical jet stream. A simple but convincing set of idealized simulations are run to explore the role of turning latitudes and PV gradients for wave propagability. It is a joy to read this well organized and clear paper. Two theoretical concepts are introduced and compared against the model results. The limitations of the model simulations are discussed. I recommend acceptance after minor revisions.

Minor points:

L2 which: change to that done

L28 suggest to replace what aspect with which properties done

L35 This is an aspect of the QR theory that is also unclear to me. In my understanding an interference of the wave with itself is not needed for QR. Rather the interference with the forced signal is crucial. However, if the wave is circumpolar the interference should be constructive. See also: https://link.springer.com/article/10.1007/s40641- 019-00150-x

Both reviewers ask a very similar question, therefore I chose to give a rather extended reply.

The phenomenon of resonance is well known in classical mechanics. Quite generally it applies to a linear system, where the full solution is a superposition of a forced (inhomogeneous) solution with a free (unforced, homogeneous) solution. The latter can be adjusted in order to satisfy initial and boundary conditions, if present. In this theory, the phenomenon of **resonance is a property of the stationary forced solution**, namely that it becomes infinite (in case of no damping) if the resonance criterion is met.

This general concept of resonance has a very clean analog in the theory of Rossby waves forced by stationary orography in a beta-channel (Held, 1983) or on the sphere (Haurwitz, 1940). **The resonance criterion** for the case of these Rossby waves says that the wavenumber of the forcing must be equal to the stationary wavenumber, and the latter is defined as the wavenumber that a free solution needs to have in order to be stationary.

In reality, this solution with infinite amplitude is never realized for two reasons. Either the system contains some damping, which regularizes the problem such that instead of an infinity the system only shows a large peak if the resonance criterion is met. Alternatively, assuming no damping and starting from a finite initial condition, the amplitude of the response grows linearly with time, which means that there is no stationary solution at all (which, in turn, makes the search for a stationary solution ill-posed). Obviously, in any real system such a gradual (linear) growth leads to violation of some of the underlying assumptions at some

point, whereafter the whole model (and in particular the assumption of linearity) turns moot.

I carefully read the article by Petoukhov et al. (2013), but I admit that I had a hard time to understand what exactly they mean by "quasi-resonance". Do they mean that in real geophysical systems there is always some amount of damping such that one only realizes the damped version of resonance? Yet, I am sure that they are refering to the framework of linear theory, since the basis of their analysis is the linearized PV equation (their equation (1)). I am also sure that they are talking about the classical mechanism of resonance as sketched above, since their key result (given in their equation (3)) relates the amplitude of the stationary solution to an expression on the right hand side which is exactly what Charney and Eliassen (1949, their equation (38)) or Held (1983, his equation (6.7)) have in their discussion of resonant Rossby waves in a zonal channel: The numerator essentially contains the amplitude of the forcing ($\tilde{A}_m^{\mathrm{ort}}$ in the terminology of Petoukhov et al. 2013) and the denominator contains the sum of two terms, namely the difference between two total wavenumbers squared plus a second term representing the damping processes. The key to resonance is that the denominator may become zero in the undamped case, if and only if these two wavenumbers are identical and, hence, their difference becomes zero.

**In no case I think that quasi-resonance can be considered to be the superposition of a free wave with a forced wave**, because (as stated above) resonance is a property of the forced solution alone. In a linear system one can always superpose a solution from the homogenous equation, but this cannot possibly lead to infinite or linearly growing amplitudes in the sense of resonance. The key difference between linear superposition of two solutions and resonance is that in the former case the amplitudes can only add (in case of constructive interference) and therefore the resulting total amplitude remains finite, while in the latter case the amplitude may become infinite (in case of resonance). In this sense resonance is "infintely more powerful" than the superposition of (or the interference between) two waves, and for this reason I believe that these two terms should be strictly distinguished from each other. It also transpires that **the sentence "a free wave _resonates with_ a starionary (or forced) wave" does not make sense**, because (1) resonance (as explained above) is a property of the forced solution alone and (2) an "interaction" between two waves can never lead to infinite amplitudes.

The general concept of resonance indicates that the linear increase of amplitude (in the case of resonance) is due to a constant input of "energy" into the system, which is allowed to "accumulate" infintely (in the case of no damping). Applying this concept to Rossby waves, the continued accumulation of wave amplitude occurs through constructive interference of a forced wave that travels around the globe in the zonal direction and then superposes with itself, and assuming that this process repeats itself again and again. This mechanism allows the wave amplitude to grow continuously and reach infinity for $t \to \infty$, which is the hall-mark of resonance in an initial value problem. Obviously, cicumglobal waves only lead to resonance as described above if the resonance criterion is met; if the resonance criterion is not met, the interference accumulates amplitude only to a certain extent and thereafter subsequent contributions add up to zero in a bunch-wise fashion. Another indication for the fact that resonance requires circumglobal waves can be found in the analysis in Charney and Eliassen (1949), where they derive the Green's function for the forced problem, i.e., the solution for a very sharp $\delta$-like mountain (their equation (38) and subsequent unlabeled equations). Their solution clearly shows that an infinite response is only obtained if and to the extent that the damping is so small that the wave can travel around the globe and "reconnects" with itself at the location of the mountain (see also their Fig. 2 illustrating this phenomenon).

In summary: yes, I believe that **resonance in the current setup requires cirumglobal Rossby waves as a prerequisite**.

L50 This statement is unclear for me.

The text was probably too terse and misleading. The other reviewer pointed to the same issue. I simply removed the sentence, as it is not really necessary at this point.

L54ff This is slightly confusing since one scale is in the meridional direction and the other scale in the longitudinal direction.

Well, when defining spatial scales, these do need to refer to the same direction, right? Here, the scale $\Delta_{bg}$ always refers to the meridional direction, while the scale of the wave $\Delta_w$ is best obtained by using the wavelength along the wavevector $\mathbf{k} = (k, l)$, and the latter may be in any direction.

L91 why is vf set to zero?

I chose to use a purely zonal flow for the pseudo-orographic forcing term. After all, the exact form of this forcing term is irrelevant for the purpose of this paper, so I preferred to make it as simple as possible. The location of the forcing is later depicted schematically in Fig. 3 (see the figure caption).

L90ff A schematic depiction of the forcing could help the reader.

The location of the forcing is schematically depicted in Fig. 3 through a yellow circle, thus clarifying the location of the forcing.

L137 delete however done

L158 void: change to devoid done

L168 could you add the equation of motion for the convenience of the reader?

That's a good point: the equation of motion was introduced earlier in section 2, and in the revised version I refer to this equation at this point. The new text reads: "First, we linearize the model equation (4) about the background state $q_0(\phi)$ ...."

L203 weak: change to small done

L217 ON the other hand: change to in contrast done

L220 which: change to: that done

L227 turning latitudes done

L292 On the other hand: change to in contrast done

L310 How independent are the jet width and the jet strength? Can these two parameters be varied completely independently?

Yes, it is possible to vary independently the jet strength and the jet width. Another option would be to prescribe latitudinal profiles of PV and (through PV inversion) deduce the associated profile of the zonal wind. However, I quite deliberately chose to specify the jet in

terms of the wind, because it facilitates the systematic sensitivity study regarding jet strength and jet width. In this respect I simply follow the lead of Manola et al. (2013).

L345 behavior done

L405 This can also interpreted from a PV anomaly point of view. The far field effect of PV anomalies would allow waves to cross areas of weaker gradients and "reinitate" a wave on a downstream waveguide.

This is a very interesting remark, thank you for pointing this out to me! I extended the text to make clear that the non-locality mentioned here is similarly borne out in the framework of PV thinking. It means that the WKB solution can only be a good approximation if and to the extent that the scale of the response from PV inversion is much smaller than the scale on which the background flow varies. This remark is particularly pertinent in the current context because in (1) PV thinking is not limited to linear perturbations, and (2) PV thinking is exact in the framework of the barotropic model.

The revised text reads: "Incidentally we note that a similar kind of non-localness is obtained in the framework of PV thinking due to the elliptic nature of the equation for PV inversion (Hoskins et al., 1985)."

L424 maybe add the equation for the group velocity

I prefer not to write down here the formula for the group velocity, because it is not needed in the remaining part of the paper. I simply assume that the reader has a broad idea about the concept of a group velocity, and that is all that's needed at this point.

**Reviewer 2 (Tim Woollings)**

This is a very nice paper which provides useful analysis and insight into the concept of waveguiding. I suggest acceptance subject to minor revisions.

One general query I had is whether any comment can be made on how much the results might apply to non-stationary situations. The analysis focuses on stationary waves but it seems the mechanisms should be similar? The O'Reilly et al case is an interesting one in this regard, with an initial transient development with non-zero phase speed acting to set up the 'circumglobal' stationary pattern.

Quite deliberately I focus in this paper on the stationary situation, because only stationary wave fields are likely to result in prolonged extreme weather situations. In this respect I follow the lead of numerous previous papers which used ray tracing concepts in order to understand the stationary eddy field. Extension to non-stationary situations would open a considerable amount of additional degrees of freedom, which in the present context is considered as unnecessary.

I wonder whether the discussion on ray tracing should be broadened a little. Of course there are many issues with a literal interpretation of this, as the author shows very convincingly, but the results here are quite specific to a zonal waveguiding situation. More broadly, ray tracing theory has some predictive power to suggest wave paths resulting from vorticity sources, for example, which just a quantification of the PV gradient doesn't. (Though this application

also has limitations, and should in general not be relied upon in isolation.) Essentially, the theory also suggests a refraction towards higher values of $K_s^2$, not just the simplistic turning latitude approach taken here.

Admittedly, the interpretation of ray tracing in my text is quite literal and I agree that a somewhat broader interpretation allows some non-trivial insight. In fact, I do mention this in my concluding section, where I reflect on the predictive power of the refractive index in, for instance, the work of Hoskins and Ambrizzi (1993), and where I say that with a somewhat broader interpretation some predictive power remains. However, it is exactly the literal interpretation that was repeatedly used in recent publications in connection with arguments of waveguidability, and this motivated me to investigate in more detail such more literal interpretation in the present paper.

My only real concern with the paper (and a minor one) is the use of just one measure of waves, named epsilon here (eq 33). In some applications, the approach is to define a conservative wave activity, by dividing the enstrophy by the mean PV gradient. This is probably not important, but I just wonder if this would alter some of the interpretation of $K_s$ - ie could $K_s$ have a better relationship to the waveguidability W if a different measure of wave activity was used?

The reviewer raises the important issue of the sensitivity of my results with respect to the waviness metric used. In fact, originally I was trying to use linear local wave activity

$$\mathcal{A} = \frac{(q^*)^2}{2\, dq_0/dy} \tag{1}$$

as a measure for waviness. However, this quantity becomes singular at those latitudes where the meridional gradient of basic state PV, $dq_0/dy$, is zero, and this does happen in all my background states that have a jet stronger than $U_J > 10$ m/s. For this reason, such a local wave activity metric turned out to be unsuitable for my purposes.

Nevertheless it is important to get a feeling for the sensitivity of the results with respect to the particular waviness metric chosen. I, therefore, repeated the calculations, but using eddy kinetic energy

$$\mathcal{K}(\lambda, \phi) = \frac{1}{2}\left\{(\bar{u}^*)^2 + (\bar{v}^*)^2\right\}\ , \tag{2}$$

as an alternative metric instead of eddy enstrophy. The corresponding dependence of waveguidability $\mathcal{W}$ on jet amplitude and jet width is very similar to the behavior shown in Fig. 6 of the paper, suggesting that my results do not sensitively depend on the metric chosen to quantify waviness. For illustration, I show the results for both metrics in Fig. 1 of this reply document: the difference between the red and the green markers is, indeed, small and irrelevant regarding the interpretations that I refer to in the paper.

In revising the manuscript, I added a paragraph in section 4.2 in which I now mention the runs with this alternative metric $\mathcal{K}$ and say that the results are essentially unchanged.

Minor queries;

line 35: Does quasi-resonance concern the resonance of the wave with itself, as suggested here, or the resonance of forced and free waves?

For the interpretation of quasi-resonance in terms of superposition of waves: see my extensive remarks in reply to reviewer 1.

[Figure]

**Figure 1:** Dependence of waveguidability $\mathcal{W}$ on jet strength $U_J$ (for fixed jet width $\sigma_J = 5$ deg). The red markers represent eddy enstrophy as waviness metric, the green markers represent eddy kinetic energy as waviness metric.

line 50: not sure this is right. In ray tracing you have to specify the zonal wavenumber, which is a property of the wave.

The reviewer is right (and the other reviewer had a similar issue here). I simply omitted this sentence, as it is not really necessary here.

line 55: clarify the length scales in this paragraph

Indeed, this section was not very clear. I enhanced the text in order to clarify. The revised text reads: "For instance, in the analysis of Kornhuber et al. (2016), the two turning latitudes are separated by about 10 deg $\approx$ 1000 km (see their Fig. 1); assuming that this distance corresponds to one half wavelength in the meridional direction and that the associated scale is given by the wavelength divided by $2\pi$, one obtains $\Delta_{bg} \approx 320$ km; but at the same time the zonal wavelength of the waves in question (wavenumber 7 at midlatitudes) is on the order of 4000 km, which means that $\Delta_w \approx 640$ km. In this situation, the relation (1) is badly violated and one would not expect WKB theory to be applicable at all."

I didn't find fig 1 very convincing. It seems there is a lot of transient activity which has to be averaged out, though I suppose this is not unlike reality. Are the magnitudes in fig 1 weaker than those in fig 3b?

First of all, I added a color bar to Fig. 1, which was missing in the original version. This color bar has the same scaling as the color bar in Fig. 3, which allows one to estimate the relative strength of the fields plotted. The reason why I want to show Fig. 1 is that it illustrates the kind of transients that one obtains from barotropic instability in the case of a strong and narrow jet. As can be seen in this Hovmoeller diagram, after some time the transients are essentially sine-shaped modes with a rather large eastward phase velocity. These transient modes make it necessary to consider a time average rather than the field itself. However, barotropic instability only occurs for rather strong and narrow jets; in other words, for most of my background states the flow is stable and the transients die out quickly leaving a stationary flow field.

lines 246-251: These sentences seem a bit out of place in a paragraph on solid body rotation. The focus is exclusively on zonal PV gradients - can the concept be extended to relate to meridional gradients and wave propagation.

This is a good point. I reworded parts of this paragraph to make clear that the addition of a jet may lead to a second turning latitude. The middle part of the paragraph now reads: " ..... In addition, $\hat{K}_s^2$ goes to zero at both poles such that at some point the expression on the right hand side of (29) turns negative leading to a turning latitude close to the pole. The crucial question in our context is whether the addition of a jet leads to a second (more equatorward) turning latitude which prevents the general equatorward refraction and forces the ray path back into the zonal direction. The latter scenario would be equivalent to a perfect zonal waveguide....."

l267-9: I'm not sure this means much - surely this is the only thing that can happen in that model?

Possibly the word "model" was misleading in this context, I changed it to "framework". What I refer to here is the idealization of the reality by a step-wise PV distribution with a PV front. The revised text reads: " ...It transpires that in this framework a zonal waveguide is present to the extent that the PV distribution has a zonally oriented sharp PV gradient separating two regions with near-homogeneous PV on both sides of the sharp gradient. ..."

l306: I would have thought it depends strongly, not slightly, on the choice of down-stream longitudes.

Well, yes, if you change the downstream longitude by a lot, the result changes a lot. The text was reworded to avoid any misinterpretation. The revised text reads:" ... it is not exact because it depends on model details like, e.g., the exact value of the damping parameter $\lambda_r$, the choice of the boundaries for the downstream region $[\lambda_1, \lambda_2]$, and the definition of "Northern midlatitudes" $[\phi_1, \phi_2]$. Nevertheless, this diagnostic is appropriate for our purpose, because we will only study relative changes of $\mathcal{W}$ as we change the background flow."

I found the argument around line 420 a bit hard to follow. Presumably a key step is that the linearisation can be excluded as a reason, given fig 10?

This was probably a little too terse in the original version. The entire argument assumes that linear theory is applicable to a reasonable approximation (as shown in Fig. 10), so this is not the point here. Rather, the point is that in the undamped limit the only parameter left in the equation, i.e., in the new equation (47), is $K_s$, suggesting that the solution can only depend on $K_s$. As a consequence, I would expect definitely a better correlation with $K_s$ in this (hypothetical) case than with $\beta_M$. Now, the numerical experiments suggest exactly the opposite, i.e., better correlation with $\beta_M$ than with $K_s$. The conclusion is that we cannot be close to this hypothetical undamped limit, or (in other words) the fact that I included dissipation (in addition to forcing) in my experiments must be crucial for the results obtained.

The revised text reads: "It implies that the solution of this equation can only depend on the combination of background fields contained in $\hat{K}_s^2$, and this should lead to a better correlation of waveguidability with $\hat{K}_s^2$ than with $\beta_M$. However, this is in conflict with our numerical results from section 4.2, which show a better correlation of waveguidability with $\beta_M$ than with $\hat{K}_s$. The only possible conclusion is that our results about waveguidability and their dependence on the background flow are essentially determined by the forced-dissipative

nature of our model configuration."

the discussion at the end of section 5 is interesting. So would the author expect $K_s$ to be more useful in the limit of no damping? Are the experiments sensitive to the damping?

Yes, I would expect that in the undamped limit the solution only depends on $K_s$, for exactly the reason mentioned in section 5 (in connection with the [new] equation (47)). However, it seems to me that this cannot be tested in any straightforward manner with the current model configuration. The reason is that the "result" of any of my model integrations is one value for waveguidability each. However, this requires (1) that the damping must not be too strong (because otherwise the wave signal is not able to travel far enough downstream to allow my kind of analysis), and (2) that the damping must not be too weak (because otherwise one would obtain circumglobal waves and, hence, interference phenomena which would prevent any straighforward interpretation in terms of waveguidability). The bottom line is that in order to diagnose waveguidability one has to more or less tune the value for $\lambda_r$ to be right inbetween the above two options of (1) too strong and (2) too weak; the value chosen in my experiments corresponds to such a tuned value for $\lambda_r$. For this reason, the obvious sensitivity experiment (namely to test the sensitivity of $\mathcal{W}$ with respect to $\lambda_r$) is not possible in any straightforward manner.

typos - l45: straddled - search for this - l59: midlatitude - l74: The key ? - l245: poles -l266: exponentially - l292: midlatitudes all corrected

**References**

Charney, J. G. and A. Eliassen, 1949: A numerical method for predicting the perturbations of the middle latitude westerlies. *Tellus*, **1**, 38–54.

Haurwitz, B., 1940: The motion of atmospheric disturbances on the spherical earth. *J. Mar. Res.*, **3**, 254–267.

Held, I. M., 1983: Stationary and quasi-stationary eddies in the extratropical troposphere: theory. In B. J. Hoskins and R. P. Pearce, editors, *Large Scale Dynamical Processes*, pp. 127–168. Academic Press.

Hoskins, B. J. and T. Ambrizzi, 1993: Rossby wave propagation on a realistic longitudinally varying flow. *J. Atmos. Sci.*, **50**, 1661–1671.

Hoskins, B. J., M. E. McIntyre, and A. W. Robertson, 1985: On the use and significance of isentropic potential vorticity maps. *Quart. J. Roy. Met. Soc.*, **111**, 877–946.

Manola, I., F. Selten, H. de Vries, and W. Hazeleger, 2013: "Waveguidability" of idealized jets. *J. Geophys. Res.*, **118**, 10,432–10,440, doi:10.1002/jgrd.50,758.

Petoukhov, V., S. Rahmstorf, S. Petri, and H.-J. Schellnhuber, 2013: Quasiresonant amplification of planetary waves and recent Northern Hemisphere weather extremes. *Proceedings of the National Academy of Sciences*, **(doi:10.1073/pnas.1222000110)**, 1–6.